# Bone marrow derived mast cells injected into the osteoarthritic knee joints of mice induced by sodium monoiodoacetate enhanced spontaneous pain through activation of PAR2 and action of extracellular ATP

Hiroko Habuchi[1☯], Masashi Izumi[2☯], Junpei Dan[2], Takahiro Ushida[1], Masahiko Ikeuchi[2], Kosei Takeuchi[3], Osami Habuchi[1]*

1 Multidisciplinary Pain Center, Aichi Medical University, Nagakute, Aichi, Japan, 2 Department of Orthopedic Surgery, Kochi Medical School, Kochi University, Okochokohasu, Nankoku, Kochi, Japan, 3 Department of Medical Cell Biology, School of Medicine, Aichi Medical University, Nagakute, Aichi, Japan

☯ These authors contributed equally to this work.
* ohabuchi@aichi-med-u.ac.jp

**Data Availability Statement:** All relevant data are within the manuscript.

## Abstract

Conditions that resemble osteoarthritis (OA) were produced by injection of sodium monoiodoacetate (MIA) into the knee joints of mice. Bone marrow derived mast cells (BMMCs) injected into the OA knee joints enhanced spontaneous pain. Since no spontaneous pain was observed when BMMCs were injected into the knee joints of control mice that had not been treated with MIA, BMMCs should be activated within the OA knee joints and release some pain-inducible factors. Protease activated receptor-2 (PAR2) antagonist (FSLLRY-NH$_2$) almost abolished the pain-enhancing effects of BMMCs injected into the OA knee joints, suggesting that tryptase, a mast cell protease that is capable of activating PAR2, should be released from the injected BMMCs and enhance pain through activation of PAR2. When PAR2 agonist (SLIGKV-NH$_2$) instead of BMMCs was injected into the OA knee joints, it was also enhanced pain. Apyrase, an ATP degrading enzyme, injected into the OA knee joints before BMMCs suppressed the pain enhanced by BMMCs. We showed that purinoceptors (P2X4 and P2X7) were expressed in BMMCs and that extracellular ATP stimulated the release of tryptase from BMMCs. These observations suggest that ATP may stimulate degranulation of BMMCs and thereby enhanced pain. BMMCs injected into the OA knee joints stimulated expression of IL-1β, IL-6, TNF-α, CCL2, and MMP9 genes in the infrapatellar fat pads, and PAR2 antagonist suppressed the stimulatory effects of BMMCs. Our study suggests that intermittent pain frequently observed in OA knee joints may be due, at least partly, to mast cells through activation of PAR2 and action of ATP, and that intraarticular injection of BMMCs into the OA knee joints may provide a useful experimental system for investigating molecular mechanisms by which pain is induced in OA knee joints.

**Funding:** This work was supported by JSPS Grant-in-Aid for Scientific Research JP16K08999 (https://www.jsps.go.jp/) (2016) to O.H. and Research Grant from Nakatomi Foundation (2017). (https://www.nakatomi.or.jp/) to M.I. The funders had no role in study design, data collection and analysis, decision to publish, or preparation of the manuscript.

## Introduction

Osteoarthritis (OA) is the most prevalent chronic joint disease and is a leading cause of impairment of mobility in the elderly population. Pain is the predominant symptom of OA; however, there is a poor correlation between the severity of disease based on radiography and symptoms of pain [1]. It is important to clear the cause and mechanisms of pain in OA for developing better therapies to help reduce symptoms and improve function. Mast cells and their mediators are known to exist in synovium and synovial fluids of OA patients and are suggested to be involved in pathogenesis of OA [2–7]. Mast cells are also proposed to be involved in the pain generation in OA [8]. Synovial tryptase derived from the activated mast cells showed significantly higher activity in OA patients than in control subjects [9]. Tryptase could activate protease activated receptor 2 (PAR2) on the plasma membrane of various cells and the activation of PAR2 was reported to be involved in the development of OA and pain generation [10, 11]. Extracellular ATP concentration in the synovial fluid has been reported to be increased in OA [12], and a clinical association between OA-associated knee pain and ATP levels in synovial fluid has been shown [13]. ATP has been reported to stimulate degranulation of mast cells [14, 15] and behave as a chemoattractant for mast cells [16].

Bone marrow derived mast cells (BMMCs) are obtained when bone marrow cells are cultured in the presence of interleukin-3 [17]. BMMCs are thought to be immature type mast cells and differentiate to both connective tissue type and mucosal type mast cells under culture conditions [18, 19] or by transfusion into genetically mast cell-deficient mice [20]. In mice that genetically lack mast cells, OA-related pathology was reduced, and transfusion of BMMCs into these mice reversed the relative protection conferred by mast cell deficiency [21]. Intraarticular injection of sodium monoiodoacetate (MIA) in rats or mice disrupts chondrocyte metabolism, leading to cartilage degeneration and subsequent subchondral bone alterations. These changes observed in the joint tissues resemble pathology in human OA [22–24], and this model has been used for studying pathogenesis of OA pain [25–27] and for therapeutic approach of OA by pharmaceutical agents [28]. Recently, we found that BMMCs elicited spontaneous pain when injected into the OA knee joints induced by MIA. In this repot, we investigated molecular mechanism by which BMMCs enhanced spontaneous pain in the OA knee joints, and found that PAR2 and extracellular ATP are possibly involved in the generation of the spontaneous pain.

## Materials and methods

### Materials

The following commercial materials were used: Alexa Fluor 546 conjugated anti-rabbit IgG antibody and Alexa Fluor 488-conjugated anti-goat IgG antibodies were from Thermo Fisher Scientific (Yokohama, Japan); anti-mast cell tryptase (V-13) antibody against mMCP-6 was from Santa Cruz Biotechnology (Santa Cruz, CA); anti-PAR2 antibody was from Bioss Inc. (Woburn, MA); anti-NF-H antibody was from Gene Tex Inc. (Irvine, CA); anti-GFP antibody was from Novus Biologicals (Centennial, CO); PAR2 antagonist peptide (FSLLRY-NH$_2$) (TOCRIS), PAR2 agonist peptide (SLIGKV-NH$_2$) (TOCRIS) and ATP Detection Assay Kit (Cayman Chemicals) were from Funakoshi Co., Ltd. (Tokyo, Japan); WEHI-3 cells were from American Type Culture Collection; RNA extraction kit (Thermo Fisher Scientific) and real time PCR kit (TAKARA) were from Takara Bio Inc. (Kusatsu, Japan); tryptase substrate (S-2288) was from Chromogenix (Milano, Italy); ATP, sodium monoiodoacetate (MIA) and apyrase were from Sigma-Aldrich Japan (Tokyo, Japan).

## Experimental design for injection of various materials and observation of pain-related behavior

Mice used in this study were 8 weeks old male C57Bl/6J (Japan SLC, Inc., Kotocho, Hamamatsu, Japan). Mice were maintained in12-hour light/12-hour dark cycles with free access to food and water in a pathogen-free room at the Laboratory Animal Research Center, Aichi Medical University. They were distributed into 8 groups (A to G). Each group included five to six mice for one experiment. Experimental design was shown in Fig 1. Injection of various materials into the right knee joints was carried out using 30G needles connected to micro syringes through polyethylene tube under anesthesia by isoflurane (2%, 1.0 l/min). According to the schema indicated in Fig 1, various materials were injected as follow: 1 mg MIA in 20 μl PBS, 20 μl PBS, 1 x $10^6$ BMMCs in 20 μl PBS, 1 x $10^6$ BMMCs plus 10 nmol PAR2 antagonist (FSLLRY-NH$_2$) in 20 μl PBS, 20 nmol PAR2 agonist (SLIGKV-NH$_2$) in 20 μl PBS, and 0.5 unit apyrase in 10 μl PBS. Pain-related mouse behavior was observed on day 0, day 6 day 13, day 15 and day 20. All animal procedures were approved by Animal Research Committee of Aichi Medical University (Approved number: 2017–26).

## Observation and quantification of spontaneous pain-related behavior

When mice were allowed to move freely in the cage, they occasionally rear up with their front limbs on the wall of cage (Fig 2a). When MIA was injected to the right knee joint, mice sometimes stood by the left hind limb alone (Fig 2b). Such behaviors were thought to protect the

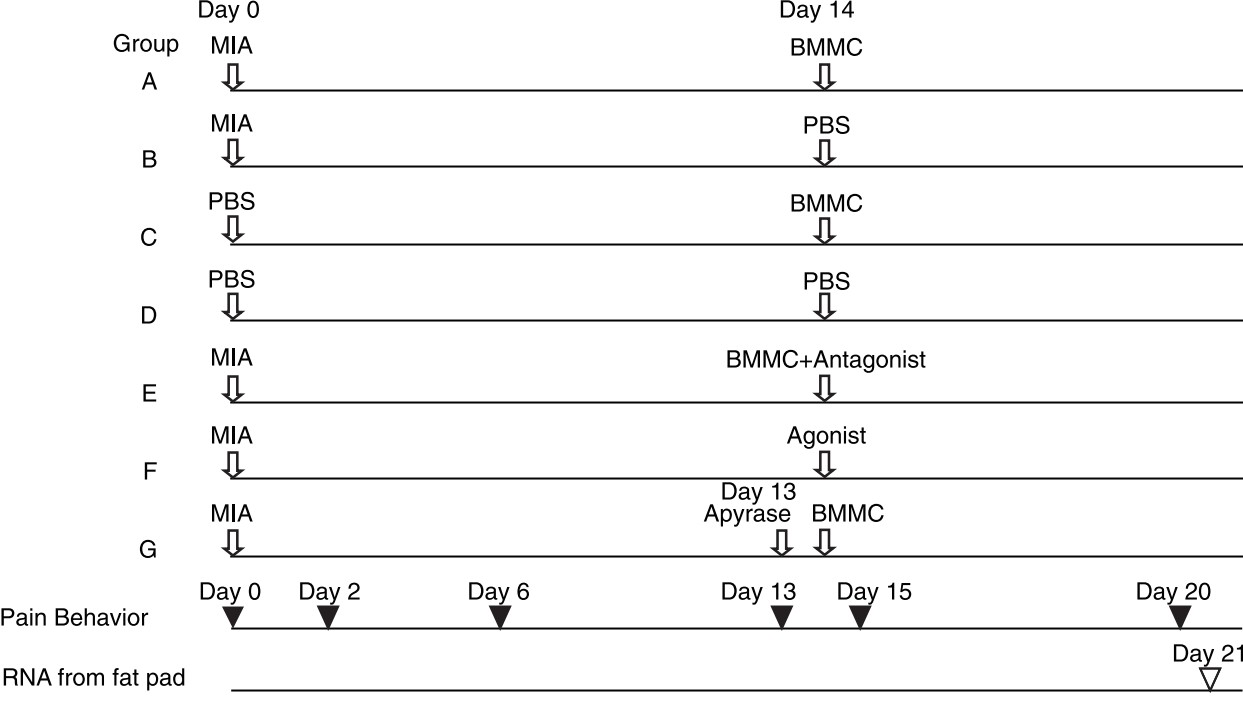

**Fig 1. Design of experiments.** Group A: MIA was injected on day 0 and BMMCs were injected on day 14; group B: MIA was injected on day 0 and PBS was injected on day 14; group C: PBS was injected on day 0 and BMMCs were injected on day 14; group D: PBS was injected on day 0 and PBS was injected on day 14; group E: MIA was injected on day 0 and a mixture of BMMCs and PAR2 antagonist was injected on day 14; group F: MIA was injected on day 0 and PAR2 agonist was injected on day 14; and group G: MIA was injected on day 0, apyrase was injected on day 13 and BMMCs were injected on day 14. The days of injection of MIA, PBS, PAR2 antagonist, PAR2 agonist and apyrase were shown in open arrows. The amount and volume injected were described under Materials and methods. Pain-related behavior was observed on the days indicated by filled arrowheads. Total RNA was extracted from the infrapatellar fat pads on the day indicated by open arrowhead.

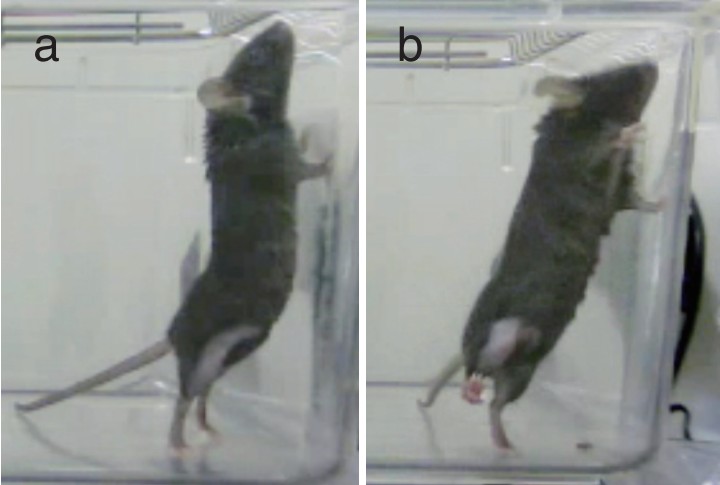

**Fig 2. Observation of pain-related behavior of mice.** When control mice were allowed to move freely in the cage, they occasionally rear up with their front limbs on the wall of cage (a). When MIA was injected to the right knee joints, mice sometimes stood with their left hind limb alone that was not treated with MIA (b).

pain that occurred in right hind limb; therefore, we quantified the pain-related behaviors using the following equation:

$$\text{Single paw standing(SPS)} = 100 \times \frac{a - b}{a + b + c}$$

where a, b and c are the number of times that mice stood with left hind limb, the number of times that mice stood with right hind limb, and the number of times that mice stood with both hind limbs, respectively, during observation period of 20 min. Observers were blind to the treatment given to each animal.

## Preparation of bone marrow derived mast cells

Bone marrow cells were obtained from the femurs of 9 to14-week-old C57BL/6 mice or EGFP transgenic mice (C57BL/6-Tg (CAG-EGFP, Japan SLC, Inc., Kotocho, Hamamatsu, Japan), and cultured in RPMI 1640 medium containing 10% fetal bovine serum, penicillin (100 units/ml), streptomycin (100 μg/ml), 2 mM L-glutamine, 0.1M nonessential amino acids, 50 μM 2-mercaptoethanol and 50% conditioned medium of WEHI-3 cells for 5-week [29, 30]. The medium was replaced every 7 days. Cells were maintained at 37 ˚C in a humidified atmosphere of 5% $CO_2$ incubator. BMMCs cultured for 5 weeks were used for the injection.

## Determination of tryptase activity released from BMMCs incubated with ATP

As an indicator of degranulation of BMMCs, we determined tryptase activity released from the cultured BMMCs. BMMCs ($1x10^6$ cells) cultured for 5 weeks were seeded in a 96-well plate and incubated with ATP (10 μM, 1 mM or 5 mM) in 100 μl PBS for 30 min at 37 ˚C. After incubation, the conditioned medium was separated from the cells by centrifugation (1500 rpm for 5min). The cell pellets were solubilized in 200 μl lysis buffer (phosphate-buffered saline containing 2 M NaCl and 0.5% Triton X-100) [30, 31]. The reaction mixtures for assaying tryptase contained, in a final volume of 120 μl, 5-μl aliquots of the lysed solution of the cell pellets or 30 μl of the conditioned medium, 20 μl of 1.8 mM aqueous solution of a chromogenic

substrate (S-2288). The reaction mixtures were incubated at 37 ˚C, and the absorbance at 405 nm was monitored with a microplate reader. The enzyme activity was determined by using a freshly extracted enzyme solution.

## Determination of ATP in the synovial fluid

Synovial fluid of the knee joint was absorbed by a strip of paper filter (2 mm x 10 mm). Volume of synovial fluid was determined by the weight increase of the paper strip. Materials absorbed to the strips were eluted with 20 μl of distilled water. Amounts of ATP in the eluates were determined using a luminescence ATP detection assay kit (Cayman Chemical Company, Ann Arbor, MI). ATP concentration was calculated from the amount of ATP and the volume of synovial fluid.

## Isolation of total RNA and determination of the level of expression of mRNAs by quantitative reverse-transcription polymerase chain reaction (qRT-PCR)

Mice were euthanized on day 21 and the infrapatellar fat pads (IFP) of the right knee joints were excised after perfusion with PBS. Total RNA was extracted from the IFP using Trizol and PureLink RNA mini kit (Thermo Fisher Scientific). Total RNA was extracted from BMMCs by the same procedures. A reverse-transcription reaction was performed using PrimeScript RT reagent kit with DNA Eraser (Takara). Quantitative RT-PCR was performed using TB Green Premix Ex TaqII (TliRNase H Plus) (Takara) and commercial specific primer pairs or custom synthesized primer pairs (P2X4 and P2X7) indicated in Table 1. Normalization of each transcript was performed using β-actin specific primer pairs (Takara). The PCR products were analyzed in real-time using ABI Prism 7000 system. More than three independent experiments were performed in triplicate to obtain the values.

## Histological analysis and immunohistochemistry of the knee joints

The right knee joints were fixed with 10% formaldehyde at 4 ˚C for 1 day and then decalcified with 0.5 M EDTA for 5 days at room temperature. The decalcified samples were embedded in paraffin according to standard histological procedures and cut into 5 μm sections. For staining of sulfated glycosaminoglycans in the cartilage, the sections were stained with 0.05% Toluidine Blue, pH 2.5. For immunostaining of GFP, mMCP6 (mouse mast cell tryptase) or neurofilament-H (NF-H), the sections were deparaffinized and then digested with 0.1% trypsin at 37˚ C for 30 min. After blocking with 5% normal goat serum (GFP, PAR2 or

**Table 1. Sequences of primers used for qPCR.**

| gene name | F primer | R primer |
|---|---|---|
| β-actin | CATCCGTAAAGACCTCTATGCCAAC | ATGGAGCCACCGATCCACA |
| CXCL2 | GCGCTGTCAATGCCTGAAG | TTTGACCGCCCTTGAGAGT |
| CCL2 | AGCAGCAGGTGTCCCAAAGA | GTGCTGAAGACCTTAGGGCAGA |
| IL-1β | AAGCTCTCCACCTCAATGGA | TGCTTGTGAGGTGCTGATGT |
| IL-6 | CCACTTCACAAGTCGGAGGCTTA | GCAAGTGCATCATCGTTGTTCATAC |
| TNF-α | ACGGCATGGATCTCAAAGAC | CGGACTCCGCAAAGTCTAAG |
| P2X4 | GCAGAAAACTTCACCCTCTTGG | AGGTAGGAGGTGGTAATGTTGG |
| P2X7 | GCAGGGGAACTCATTCTTTGTG | TCCACCCCTTTTTACAACGG |
| MMP9 | CCATGCACTGGGCTTAGATCA | GGCCTTGGGTCAGGCTTAGA |
| CGRP | TTGTCAGCATCTTGCTCCTGTACC | TTCATCTGCATATAGTCCTGCACCA |

NF-H) or 5% normal donkey serum (mMCP6), the sections were incubated with anti-GFP antibody, anti-mMCP6 antibody or anti-NF-H antibody as the first antibody at 4 ˚C over-night. After washing, sections were stained with Alexa Fluor 546-conjugated anti-rabbit IgG antibody for GFP, PAR2 and NF-H or Alexa Fluor 488-conjugated anti-goat IgG anti-body for mMCP6. The nuclei were stained with DAPI. For immunostaining of PAR2 and NF-H, trypsin digestion was omitted. Images were captured using a KEYENCE BZ-9000 microscope.

## Statistical analysis

Comparisons between groups were assessed by Mann-Whitney U test. Values of $p < 0.05$ were considered to be statistically significant.

## Results

### BMMCs injected into the MIA-induced OA knee joints enhanced spontaneous pain

When MIA was injected into the knee joints of mice, the superficial layer of cartilage was dam-aged (arrowheads in Fig 3, group A, B, E and F). Obvious pain behavior was observed on day 2 after injection of MIA, and the pain behavior gradually decreased thereafter (Fig 4, group A and group B). Intraarticular injection of BMMCs on day 14 enhanced the pain behavior again (Fig 4, group A), whereas no pain enhancement was observed when PBS was injected on the same day (Fig 4, group B). When PBS was injected on day 0 and BMMCs were injected on day 14, no pain behavior was observed (Fig 4, group C). Injection of PBS on both day 0 and day 14 caused no effects (Fig 4, group D). To confirm that injected BMMCs were retained in the joint space correctly, we injected BMMCs obtained from EGFP-transgenic mice into the knee joint on day 14 and detected the injected BMMCs by antibodies against GFP and mMCP-6 on day 15 (Fig 5). The cells that were positive to both GFP and mMCP-6 were localized near the syno-vial membrane in both group A and C, indicating that injected BMMCs were retained within the knee joints. A portion of the mMCP6-positive cells showed shapes characteristic of the degranulated BMMCs (Fig 5d, inset). These observations seem to support the idea that BMMCs retained in the joint space of the OA knee joints were activated and released some fac-tors that are capable of inducing pain. In group A (Fig 5d) and group C (Fig 5h), cells that were positive to mMCP-6 but negative to GFP were also observed. These cells are thought to be endogenous mast cells recruited to the osteoarthritic knee joint. Number of these cells in group A (Fig 5d) appeared to be larger than that in group C (Fig 5h).

### Enhancement of spontaneous pain elicited by BMMCs injected into the OA knee joints requires activation of PAR2

PAR2 is a member of PAR family belonging to G protein-coupled receptors and activated by serine protease such as mast cell tryptase. Tryptase is a major mast cell protease and released from mast cells or BMMCs on the degranulation. To show a possibility that activation of PAR2 by tryptase might be involved in the enhancement of spontaneous pain elicited by BMMCs injected into the OA knee joints, we examined the effects of an antagonist for PAR2 on the pain enhancement. BMMCs injected into the OA knee joint in the presence of PAR2 antagonist peptide (FSLLRY-NH$_2$) failed to enhance pain (Fig 6, group E), suggesting that acti-vation of PAR2 by tryptase released from BMMCs should be required for the pain enhance-ment by BMMCs. This idea seems to be supported by the observation that PAR2 agonist

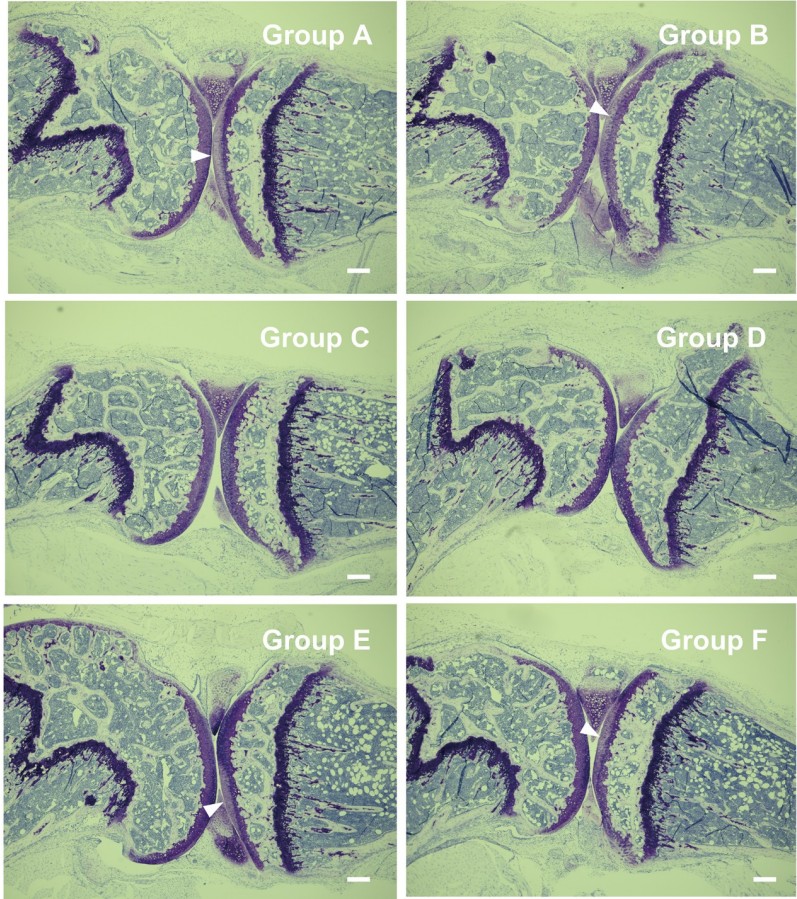

**Fig 3. Toluidine blue staining of the knee joints of mice treated with MIA.** Knee joints were excised from mice belonging to group A, B, C, D, E and F on day 21 and processed according to the methods described under Materials and methods. Original magnification was x40. Bars indicate 200 μm. Damage of cartilage was observed in group A, B, E and F (indicated by arrowheads).

(SLIGKV-NH$_2$) instead of BMMCs injected into the OA knee joints on day 14 significantly enhanced pain (Fig 7, group F).

## BMMCs injected into the OA knee joints stimulated expression of inflammation-related genes in the infrapatellar fat pads (IFP) through activation of PAR2

Expressions of various mRNAs in the IFP of mice belonging to group A (MIA was injected on day 0, and BMMCs were injected on day 14), group B (MIA was injected on day 0, and PBS was injected on day 14) and group E (MIA was injected on day 0, and BMMCs were injected on day 14 in the presence of PAR2 antagonist) were determined on day 21. Expression of IL-1β, IL-6, TNF-α, CCL2, MMP9 and CGRP in the IFP of mice belonging to group A was higher than those belonging to group B (Fig 8a). On the other hand, expression of these genes in the IFP of mice belonging to group E was lower than those belonging to group A (Fig 8b). These observations suggest that stimulation of the gene expression in the IFP by BMMCs injected into the OA knee joints may be mediated by activation of PAR2 with tryptase released from BMMCs.

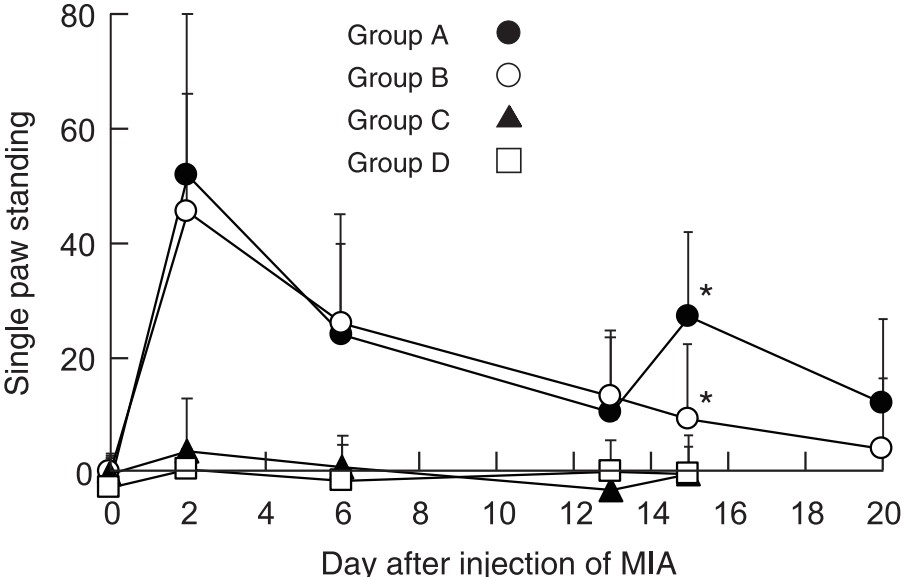

**Fig 4. Observation of pain-related behavior of mice belonging to group A, B, C, and D.** Single paw standing was calculated on day 0, 2, 6, 13, 15 and 20 according to the methods described under Materials and methods. Data from five experiments were combined. Total number of mice used were 24, 20, 8, and 15 for group A, B, C and D, respectively. SDs were indicated by bars. *p<0.001.

## PAR2 was expressed in the tissues of the OA knee joints

To examine whether PAR2 is actually expressed in the knee joint tissues, we performed immunofluorescent stain with anti PAR2 antibody. PAR2-positive signals were observed in the surface of meniscus (Fig 9a and 9b), chondrocytes of epiphyseal cartilage (Fig 9b), synovial membrane (Fig 9e), and subchondral bone (Fig 9a and 9e) of mice belonging to group A (MIA was injected on day 0, and BMMCs were injected on day 14). Intensity of the signals became weaker in mice belonging to group C (PBS was injected on day 0, and BMMCs were injected on day 14) (Fig 9c) or almost disappeared in mice belonging to group D (PBS was injected on day 0, and PBS was injected on day 14) (Fig 9d). Positive signals of PAR2 on synovial membrane of mice belonging to group A (Fig 9e) roughly overlapped with the positive signals of neurofilament H (Fig 9f).

## ATP is involved in the enhancement of spontaneous pain elicited by BMMCs injected into the OA knee joints

Extracellular ATP was reported to cause an excitation of nociceptive afferent nerves innervating normal knee joints of rats (32) and is known as a signal molecule that induces degranulation of mast cells (14). To examine a possibility that extracellular ATP might be involved in the enhancement of spontaneous pain elicited by BMMCs injected into the OA knee joints, we investigated the effects of apyrase, a nucleotidase of plant origin that selectively degrades ATP, on the pain enhancement elicited by BMMCs. When BMMCs were injected into the OA knee joints after injection of apyrase or vehicle (PBS), the pain of the mice belonging to group G (MIA was injected on day 0, apyrase was injected on day 13 and BMMCs were injected on day 14) was significantly lower than the pain of mice belonging to group A (in this experiment, MIA was injected on day 0, PBS was injected on day 13 and BMMCs were injected on day 14) (Fig 10). We confirmed that mRNAs of P2X7 and P2X4, members of cell surface receptors for ATP, were actually expressed in BMMCs (Fig 11a), and that ATP could stimulate the release of

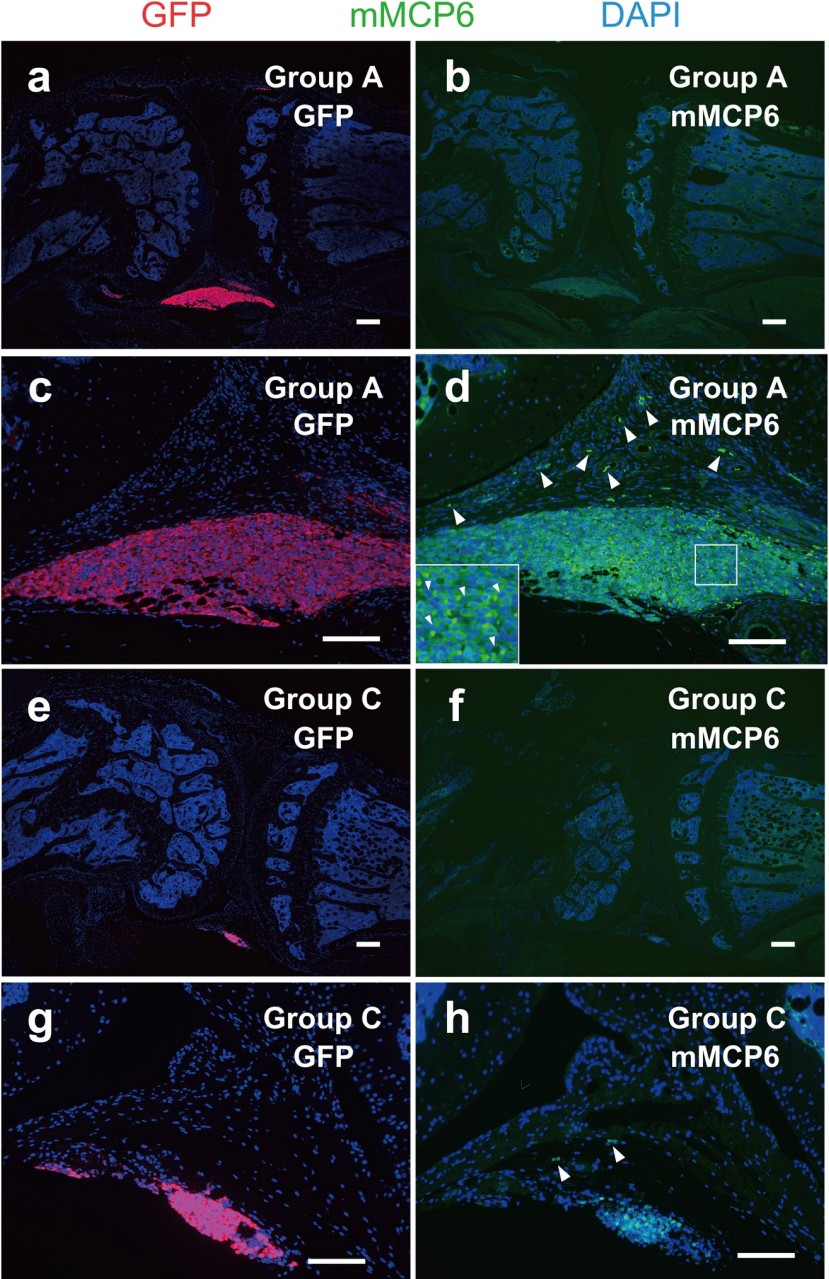

**Fig 5. Localization of GFP-labeled BMMCs in the knee joints.** Knee joints were excised from mice belonging to group A and C on day 15 (24 h after injection of GFP-labeled BMMCs). Neighboring sections were stained with anti-GFP antibody (red) (a, c, d, and g) or anti-mMCP6 antibody (green) (b, d, f and h) according to the methods described under Materials and methods. In inset of d, an enlarged image of the area surrounded by a square was shown. In this inset, degranulated BMMCs were indicated by small arrowheads. Endogenous mast cells that were positive to mMCP6 but negative to GFP were indicated by arrowheads in d and h. Original magnification was x40 (a, b, e, f) and x200 (c, d, g, h). Bars indicate 200 μm (a, b, e, f) and 100 μm (c, d, g, h).

tryptase from BMMCs in a dose-dependent manner (Fig 11b). Taken together, it may be possible that extracellular ATP is involved in the enhancement of spontaneous pain elicited by BMMCs through, at least partly, activation and degranulation of BMMCs. We also tried to show that extracellular ATP was actually increased in the synovial fluid of the OA knee joints.

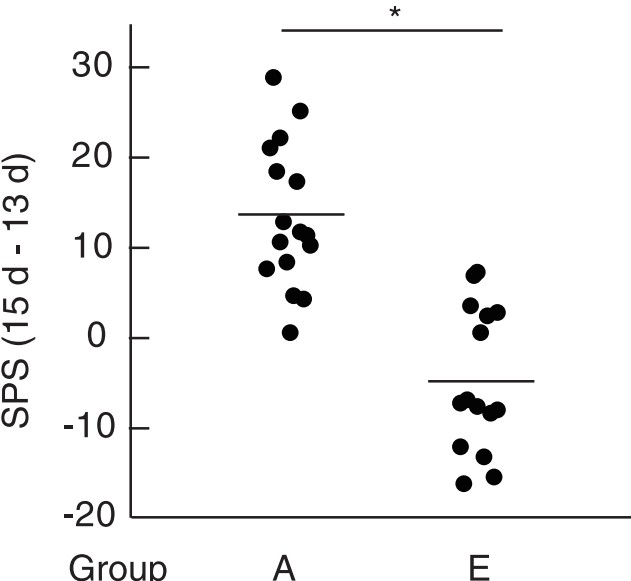

**Fig 6. Effects of PAR2 antagonist (FSLLRY-NH$_2$) on the pain-related behaviors.** Calculation of single paw standing of mice belonging to group A and E were performed according to the methods described under Materials and methods. Data from four experiments were combined. Total number of mice used were 15 and 14 for group A and group E, respectively. Each dot represents difference of single paw standing (ΔSPS) between day 15 and day 13 of a single mouse. Averages were shown by bars. * p< 0.0001.

ATP concentration in the synovial fluid of control mice was 0.40 ± 0.25 μM. On day 2 after injection of MIA, we found that volume of the synovial fluid increased about five-fold compared to that of the control mice; however, we failed to determine the ATP concentration because ATP assay was strongly interfered for unknown reasons.

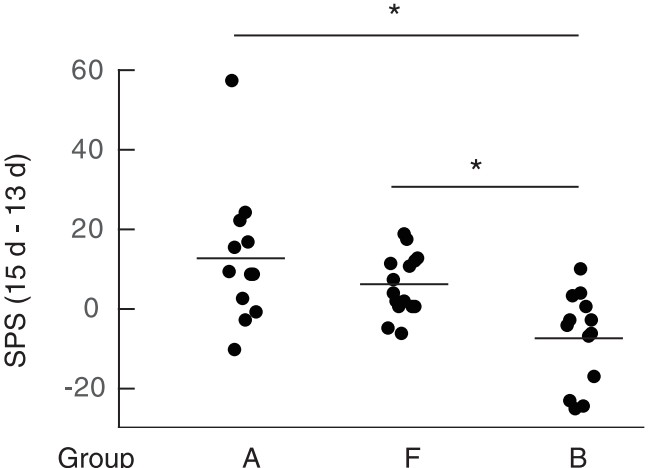

**Fig 7. Effects of PAR2 agonist (SLIGKV-NH$_2$) on the pain-related behaviors.** Calculation of single paw standing of mice belonging to group A, B and F were performed according to the methods described under Materials and methods. Data from four experiments were combined. Total number of mice used were 13, 15 and 14 for group A, group F and group B, respectively. Each dot represents difference of single paw standing (ΔSPS) between day 15 and day 13 of a single mouse. Averages were shown by bars. *p< 0.005.

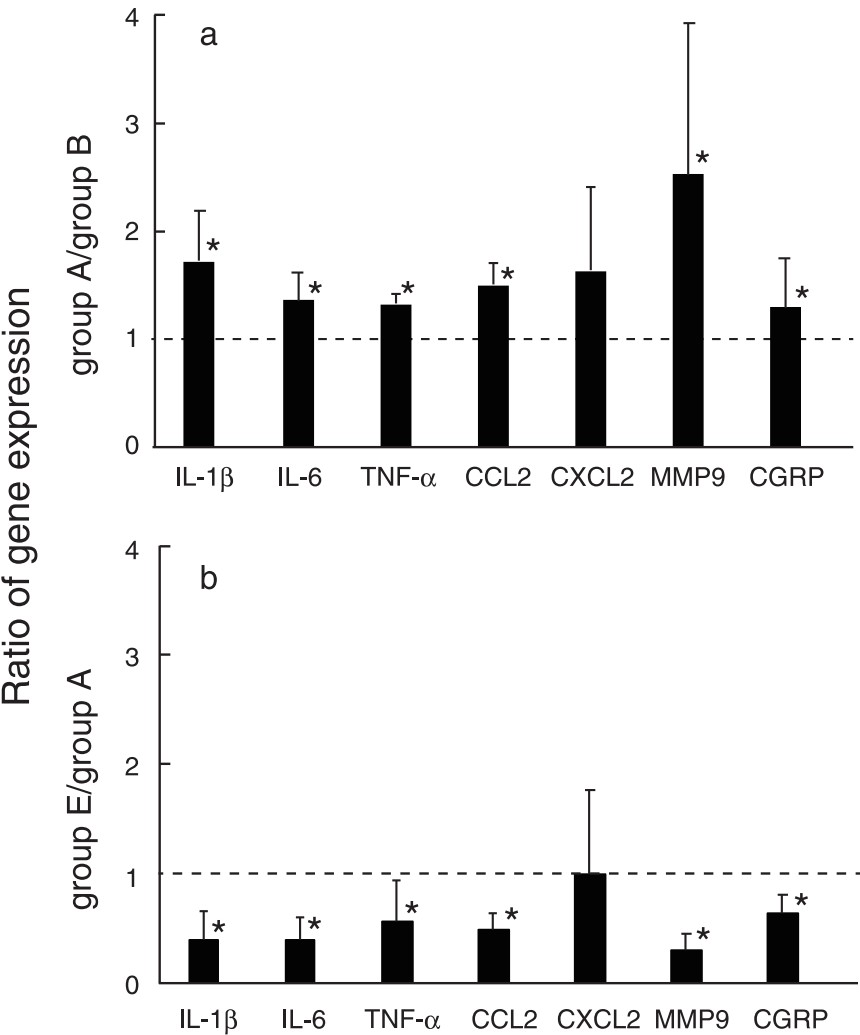

**Fig 8. Effects of BMMCs and PAR2 antagonist (FSLLRY-NH₂) on the expression of various genes in the infrapatellar fat pads of the OA knee joints.** a, Stimulation of the various gene expression by BMMCs. Height of each bar represents the ratio of the gene expression in group A to the gene expression in group B. SD was indicated above each bar. *p<0.05. b, Suppression of the gene expression by PAR2 antagonist. Height of each bar represents the ratio of the gene expression in group E to the gene expression in group A. SD was indicated above each bar. *p<0.05. Extraction of total RNA and qPCR were performed according to the methods described under Materials and methods. Data from four experiments were combined. Total number of mice used were 12 for each group. Expression levels of each mRNA were normalized with the expression level of β-actin.

## Discussion

We found that BMMCs injected into the MIA-induced OA knee joints of mice enhanced spontaneous pain (Fig 4, group A). In contrast, BMMCs injected into the control mice that had not been treated with MIA could not enhance pain at all (Fig 4, group C). These observations strongly suggest that BMMCs injected into the OA knee joints were activated in the knee joint space and released some pain-enhancing factors. This idea seems to be supported by the observation that a portion of the injected BMMCs showed shapes characteristic of degranulated mast cells when stained with anti-mMCP6 antibody (Fig 5, inset). Among these factors, we focused on tryptase, a mast cell-specific protease, because tryptase has been known to be expressed in BMMCs [29] and considered as the major signaling molecule which can regulate

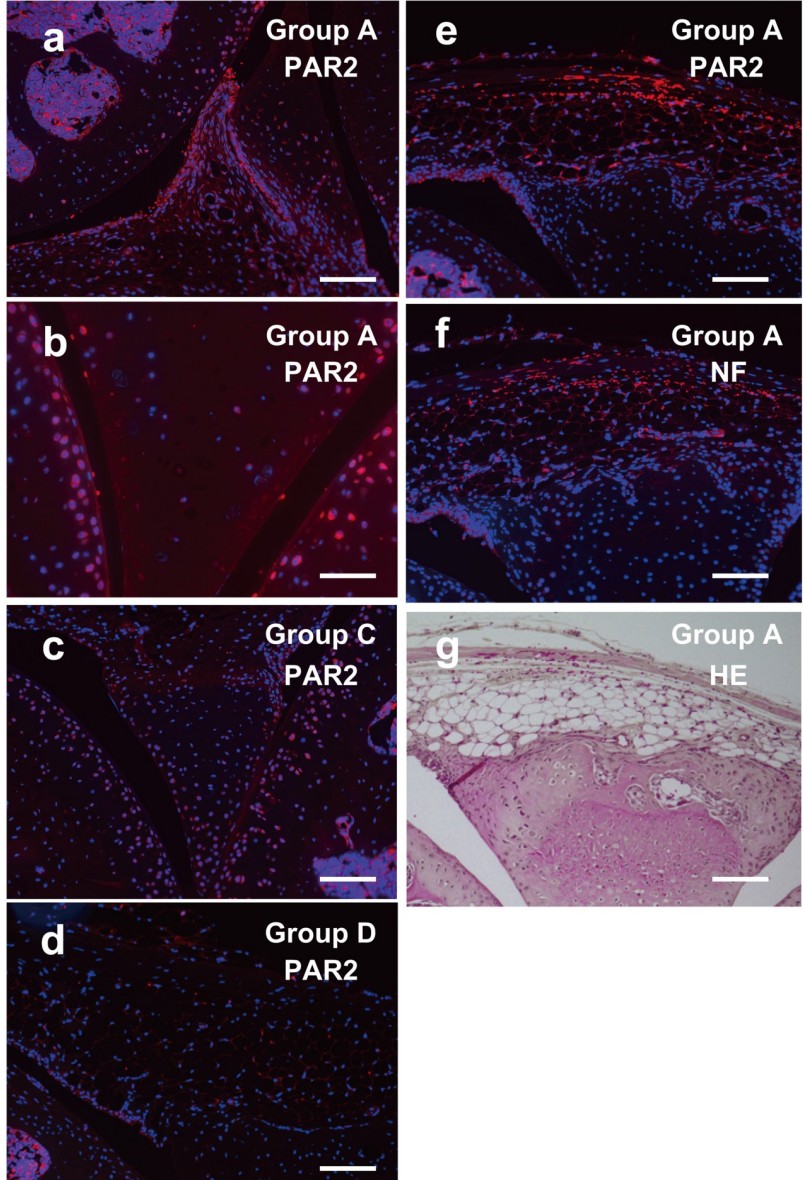

**Fig 9. Expression of PAR2 in the OA knee joints.** Knee joints from group A (a, b, e–g), group C (c) and group D (d) were excised 3 h after injection of BMMCs (a–c, e and f) or PBS (d) on day 14 and stained with anti PAR2 antibody (a–e), anti-NF-H antibody (f) or hematoxylin-eosin (g). Neighboring sections were stained in e-g. Original magnification was x200 (a, c–g) and x400 (b). Bars indicate 100 μm (a, c–g) and 50 μm (b).

nociceptive transmission through activation of PAR2 [32, 33]. We found that PAR2 antagonist reduced the pain-enhancing effects of BMMCs, suggesting that tryptase released from BMMCs on the degranulation might contribute to the pain enhancement through the activation of PAR2. Our observation seems to be well coincided with other reports in which deletion of PAR2 delayed the progression of OA and decreased pain following destabilization of the medial meniscus [10, 11, 34]. It is well known, however, that various nociceptive factors other than tryptase such as histamine are released from mast cells on the degranulation; therefore, a possibility still remains that some factors other than tryptase might contribute to the pain-

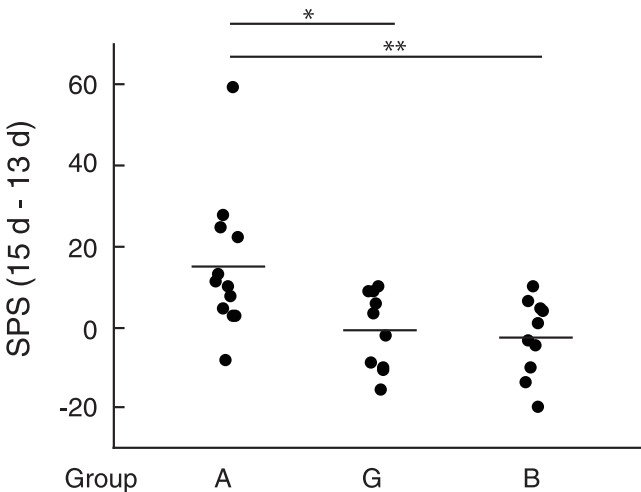

**Fig 10. Effects of apyrase on the pain-related behaviors.** Calculation of single paw standing of mice belonging to group A, B and G were performed according to the methods described under Materials and methods. In this experiment, PBS was injected instead of apyrase on day 13 in group A and B. Data from three experiments were combined. Total number of mice used were 12, 10 and 10 for group A, group G and group B, respectively. Each dot represents difference of single paw standing (ΔSPS) between day 15 and day 13 of a single mouse. Averages were shown by bars. $^*p < 0.02$, $^{**}p < 0.01$.

enhancing effects of BMMCs. Nevertheless, in the presence of PAR2 antagonist, major part of the pain enhanced by the injected BMMCs disappeared (Fig 6), indicating that tryptase should play a major role as the pain-inducing factor in our experiments. Since PAR2 was expressed in mast cells and functioned in mast cell activation [35–37], an alternative possibility could not be excluded that degranulation of BMMCs might be inhibited by PAR2 antagonist.

Our observations that apyrase reduced the pain-enhancing effects of (Fig 10) seem to support the idea that extracellular ATP, if not solely, might participate in the activation and degranulation of the injected BMMCs. ATP concentration in the synovial fluid of control mice was found to be 0.40 ± 0.25 μM. This concentration appears to be too low to activate purinoceptors, because ATP concentrations required to activate P2X4 receptor and P2X7 receptor were reported to be micromolar and millimolar range, respectively [14, 38, 39]. ATP concentrations of synovial fluid of the OA knee joints were expected to be higher than those of control mice, but we failed to determine them because ATP assay was strongly interfered. The reason why ATP assay was interfered is not clear at present but may be related to the observations that firefly luciferase was inhibited by various materials [40–42]. Since P2X receptors are expressed in the subsets of primary afferent neurons, and excitation of sensory neurons by ATP evokes a sensation of pain [43, 44], an alternative possibility still remains that ATP released to the knee joint space from the surrounding tissues or from the injected BMMCs might be degraded by apyrase and hence could not stimulate purinoceptors expressed on the sensory nerve terminals. However, pain resulting from stimulation of purinoceptors by ATP is expected to be low, if any, in our experiments, because pain enhanced by BMMCs injected into the OA knee joints was almost abolished by the addition of PAR2 antagonist.

In Fig 3, deterioration of superficial cartilage detected by toluidine blue staining was evident in mice belonging to A, B, E and F groups; however, significant difference in the degree of the deterioration was not observed among these groups. These results may be related to the clinical observation that OA pain does not necessarily correlate with the extent of joint degeneration [1].

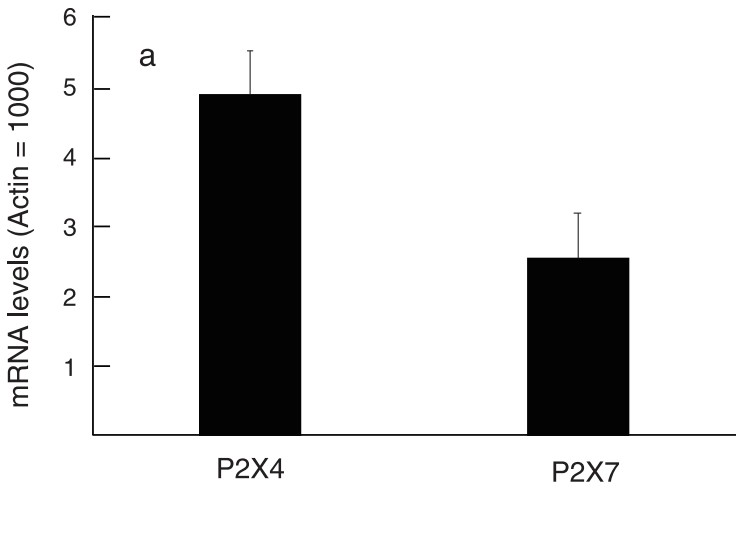

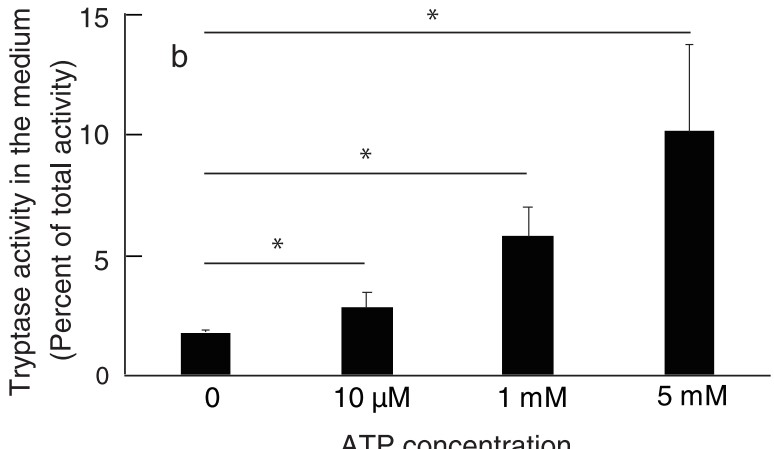

**Fig 11. Expression of mRNA of P2X4 and P2X7 in BMMC (a) and effects of ATP on the release of tryptase from BMMCs (b).** a, Height of bars represents ratios of the gene expression of P2X4 and P2X7 to the gene expression of β-actin in BMMCs. SDs were shown above bars. b, Height of bars represents the ratios of tryptase activity in the medium to the total tryptase activity (sum of the activity in the medium and the cell pellets). Four independent experiments were carried out. SDs were shown above bars. $^*$p<0.05.

Pain development in OA has been shown to be well correlated with bone marrow lesion [1, 45, 46]. In osteoarthritic subchondral bones, the most abundant infiltrating immune cells were shown to be T cell CD8, activated mast cells and activated T cell CD4 memory, while in normal synovial samples, the most abundant infiltrating immune cells were resting mast cells, monocytes, and neutrophils [47]. BMMCs injected into the OA knee joints may function in a similar manner to the abundant activated mast cells present in the bone marrow lesion.

BMMCs injected into the OA knee joints not only enhanced pain but also stimulated the gene expression of IL-1β, IL-6, TNF-α, CCL2, MMP9 and CGRP in the IFP. On the other hand, PAR2 antagonist suppressed both pain and the gene expression enhanced by BMMCs. Considering that IFP has been suggested to contribute to pain in OA knee joints [48, 49], the stimulated expression of these genes in the IFP mediated by PAR2 might contribute to the pain enhancement. But it remains to be investigated whether proteins resulting from the expression of these genes might actually be involved in the pain enhancement.

Pain behavior of mice with MIA-induced knee OA has been previously observed using von Frey filament test (27). They found that values of paw withdrawal threshold became low on day 3 after injection of MIA and were maintained at the low level for four weeks. In contrast, we found that values of single paw standing showed a peak on day 2 after injection of MIA and gradually decreased thereafter. The values increased again only after injection of BMMCs. These observations suggested that single paw standing was more suitable for detecting the effects of BMMCs on the pain occurred in the OA knee joints; therefore, we used single paw standing throughout our experiments.

Our study suggests that the intermittent pain frequently observed in OA knee joints may be due, at least partly, to mast cells through activation of PAR2 and action of ATP, and that intra-articular injection of BMMCs into the OA knee joints may provide a useful system for investigating molecular mechanism by which pain is induced in the OA knee joints.

## Conclusion

Bone marrow derived mast cells (BMMCs) injected into the MIA-induced OA knee joints enhanced spontaneous pain, but these cells showed no pain-enhancing effects when injected into the knee joints of control mice that had not been treated with MIA. PAR2 and extracellular ATP were possibly involved in the pain-enhancing effects of BMMCs.

## Acknowledgments

We thank Reika Shiraishi (Kochi Medical School) for invaluable technical assistance.

## Author Contributions

**Conceptualization:** Hiroko Habuchi, Masashi Izumi, Takahiro Ushida, Masahiko Ikeuchi, Osami Habuchi.

**Funding acquisition:** Masashi Izumi, Osami Habuchi.

**Investigation:** Hiroko Habuchi, Masashi Izumi, Junpei Dan, Osami Habuchi.

**Resources:** Kosei Takeuchi.

**Writing – original draft:** Hiroko Habuchi, Osami Habuchi.

**Writing – review & editing:** Osami Habuchi.

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
