## [Decision Letter · Decision Letter 0]

9 Mar 2021

PONE-D-21-01925

Bone marrow derived mast cells injected into the osteoarthritic knee joints of mice induced by sodium monoiodoacetate enhanced spontaneous pain through activation of PAR2 and action of extracellular ATP

PLOS ONE

Dear Dr. 

Thank you for submitting your manuscript to PLOS ONE. After careful consideration, we feel that it has merit but does not fully meet PLOS ONE’s publication criteria as it currently stands. Therefore, we invite you to submit a revised version of the manuscript that addresses the points raised during the review process.

We look forward to receiving your revised manuscript.

Kind regards,

Rosanna Di Paola, MD

Academic Editor

PLOS ONE

Journal Requirements:

Reviewers' comments:

Reviewer's Responses to Questions

**Comments to the Author**

1. Is the manuscript technically sound, and do the data support the conclusions?

Reviewer #1: Yes

Reviewer #2: Yes

2. Has the statistical analysis been performed appropriately and rigorously? 

Reviewer #1: Yes

Reviewer #2: Yes

3. Have the authors made all data underlying the findings in their manuscript fully available?

Reviewer #1: Yes

Reviewer #2: Yes

4. Is the manuscript presented in an intelligible fashion and written in standard English?

Reviewer #1: Yes

Reviewer #2: Yes

5. Review Comments to the Author

Reviewer #1: This is an interesting manuscript that addresses the role of mast cells in osteoarthritis pain in a chemically induced OA model. It has been shown before that mast cells may be involved in OA pathogenesis and pain but the detailed mechanisms are not known. The data shown in this manuscript may help us to understand the underlying mechanisms. The authors need to address the following questions.

1) The manuscript should be edited for clarity in English.

2) It seems that mast cells do not initiate joint pain but worsen joint pain in OA mice that suffer joint degeneration. In human, OA pain does not necessarily correlate with the extent of joint degeneration. The authors may want to quantify OA scores of the mice in different groups in Figure 3. This is to determine whether OA pain is related to joint degeneration in this OA mouse model.

3) The single limb standing may not be a well-established pain assessment method. Is there any literature about its efficacy in assessing OA pain? How does it compare to gait analysis or other more common pain assessment method?

4) The authors may want to add a mechanistic diagram to summarize their working hypothesis of the involvement of different joint tissues including fat pad, subchondral bone marrow, and synovium and the signaling pathways including PAR2, ATP, and inflammatory cytokines in inducing OA pain.

Reviewer #2: Hiroko Habuchi and colleagues investigated about the role of mast cells in spontaneous pain induced by injection of sodium monoiodoacetate (MIA) into the knee joints of mice.

The rational behind the study was clear and straight forward. The paper is clearly written, its original and of interest in its field, but some details are not clear.

I recommend that the paper be accepted with minor revision:

a) The authors should better emphasize the conclusions.

b) The authors should clarify how they choose the number of mice used in their study.

As to the methodology, please specify the number of animals used per each technique.

c) While many different sources are used to set up the study in the introduction, little previous evidence is stated. Incorporating comparisons with other studies would increase the strength of the paper. Please refer to doi: 10.3390/ani10101827; 10.3390/antiox9060511; 10.1186/s13075-019-2048-y

6. PLOS authors have the option to publish the peer review history of their article (what does this mean?). If published, this will include your full peer review and any attached files.

Reviewer #1: **Yes: **Qian Chen

Reviewer #2: No

---

## [Author Response · Author response to Decision Letter 0]

14 Mar 2021

Dear Dr. Rosanna Di Paola 

Academic Editor

PLOS ONE

 Thank you for your E-mail of March 9, 2021 about our manuscript entitled "Bone marrow derived mast cells injected into the osteoarthritic knee joints of mice induced by sodium monoiodoacetate enhanced spontaneous pain through activation of PAR2 and action of extracellular ATP." (PONE-D-21-01925). I have amended our manuscript as far as possible in light of the opinion of the reviewers. To answer the question raised by the reviewer, we revised the manuscript. 

Answer to the reviewer

For reviewer #1: 

1) Some descriptions in Abstract (Line 26 and Line 33-36), Introduction (Line 55-57) and Discussion (Line 360, Line 366, Line 374 and Line 380) were modified for clarity in English.

2) In Fig 3, deterioration of superficial cartilage detected by toluidine blue staining was evident in mice belonging to A, B, E and F groups; however, significant difference in the degree of the deterioration was not observed among these groups. These results may be related to the clinical observation that OA pain does not necessarily correlate with the extent of joint degeneration. We added such description in Discussion (Line 398-402).

3) In the preliminary experiments, we have also observed stance score on the basis of the previously described methods (Heilborn U, Berge OG, Arborelius L, Brodin E. Spontaneous nociceptive behaviour in female mice with Freund's complete adjuvant- and carrageenan-induced monoarthritis. Brain Res. 2007 Apr 27;1143:143-9. doi: 10.1016/j.brainres.2007.01.054), and obtained basically the similar results to those from single paw standing; however, we adopted single paw standing for quantification of pain in this paper because single paw standing gave more quantitative data than stance score. 

4) Our experiments clearly indicate that PAR2 and ATP are involved in the generation of spontaneous pain in OA; however, many other factors and different types of cells and tissues are possibly involved in elicitation of pain. We are concerned about too much simplification of the mechanistic aspect of OA pain by drawing a diagram that summarize our working hypothesis; 

therefore, we would like to omit the diagram that the reviewer has suggested. 

For reviewer #2: 

a) We described conclusion in the newly inserted section of Conclusion as follow: “Bone marrow derived mast cells (BMMCs) injected into the MIA-induced OA knee joints enhanced spontaneous pain, but these cells showed no pain-enhancing effects when injected into the knee joints of control mice that had not been treated with MIA. PAR2 and extracellular ATP were possibly involved in the pain-enhancing effects of BMMCs. “ (Line 430-434).

b) We used enough mice to obtain significant results. Number of mice used for each experiment was described in the legends of Fig 4, Fig 6, Fig 7 and Fig 10. 

c) In introduction we added one paper about the effects of a pharmaceutical agent on the pain elicited in MIA-induced OA (Ref 28) (Line 70). 

We found after submission that Ref 49 was the same as Ref 27, so we deleted Ref 49. Accordingly Ref number was modified. We modified reference format to fit it to the PLOS ONE duideline. 

I believe that modification of the manuscript made this paper more correct and better. I would like to thank the reviewers for their kind comments. I hope this revised paper is now suitable for publication in the PLOS ONE. 

Sincerely yours,

Osami Habuchi, Ph.D.

---

## [Editor Report · Decision Letter 1]

19 May 2021

Bone marrow derived mast cells injected into the osteoarthritic knee joints of mice induced by sodium monoiodoacetate enhanced spontaneous pain through activation of PAR2 and action of extracellular ATP

PONE-D-21-01925R1

Dear Dr. 

We’re pleased to inform you that your manuscript has been judged scientifically suitable for publication and will be formally accepted for publication once it meets all outstanding technical requirements.

Kind regards,

Rosanna Di Paola, MD

Academic Editor

PLOS ONE
---

## [Editor Report · Acceptance letter]

24 May 2021

PONE-D-21-01925R1 

Bone marrow derived mast cells injected into the osteoarthritic knee joints of mice induced by sodium monoiodoacetate enhanced spontaneous pain through activation of PAR2 and action of extracellular ATP 

Dear Dr. Habuchi:

I'm pleased to inform you that your manuscript has been deemed suitable for publication in PLOS ONE. Congratulations! Your manuscript is now with our production department. 

Kind regards, 

on behalf of

Dr. Rosanna Di Paola 

Academic Editor

PLOS ONE